

# A simple, sufficient, and consistent method to score the status of threats and demography of imperiled species

Jacob W. Malcom[1], Whitney M. Webber[2] and Ya-Wei Li[1]

[1] Endangered Species Conservation, Defenders of Wildlife, Washington, D.C., United States of America
[2] Department of Earth & Environmental Science, University of Pennsylvania, Philadelphia, PA, United States of America

## ABSTRACT

Managers of large, complex wildlife conservation programs need information on the conservation status of each of many species to help strategically allocate limited resources. Oversimplifying status data, however, runs the risk of missing information essential to strategic allocation. Conservation status consists of two components, the status of threats a species faces *and* the species' demographic status. Neither component alone is sufficient to characterize conservation status. Here we present a simple key for scoring threat and demographic changes for species using detailed information provided in free-form textual descriptions of conservation status. This key is easy to use (*simple*), captures the two components of conservation status without the cost of more detailed measures (*sufficient*), and can be applied by different personnel to any taxon (*consistent*). To evaluate the key's utility, we performed two analyses. First, we scored the threat and demographic status of 37 species recently recommended for reclassification under the Endangered Species Act (ESA) and 15 control species, then compared our scores to two metrics used for decision-making and reports to Congress. Second, we scored the threat and demographic status of all non-plant ESA-listed species from Florida (54 spp.), and evaluated scoring repeatability for a subset of those. While the metrics reported by the U.S. Fish and Wildlife Service (FWS) are often consistent with our scores in the first analysis, the results highlight two problems with the oversimplified metrics. First, we show that both metrics can mask underlying demographic declines or threat increases; for example, ~40% of species not recommended for reclassification had changes in threats or demography. Second, we show that neither metric is consistent with either threats or demography alone, but conflates the two. The second analysis illustrates how the scoring key can be applied to a substantial set of species to understand overall patterns of ESA implementation. The scoring repeatability analysis shows promise, but indicates thorough training will be needed to ensure consistency. We propose that large conservation programs adopt our simple scoring system for threats and demography. By doing so, program administrators will have better information to monitor program effectiveness and guide their decisions.

Corresponding author
Jacob W. Malcom,
jmalcom@defenders.org

# INTRODUCTION

The administration and monitoring of conservation programs are closely entwined. Administrators charged with conserving imperiled species must do so often under acute budget and personnel constraints (*Ferraro & Pattanayak*, *2006*). At the national and regional scales, these decision makers need to accurately evaluate hundreds or thousands of species based on their conservation status as part of their decision on how to allocate limited resources efficiently and objectively for the greatest conservation benefit (*Bottrill et al.*, *2008*; *Joseph et al.*, *2008*). At the same time, assessing the effectiveness of large, complex conservation programs is challenging because of the taxonomic breadth of species and the variety of threats they face (see *Purvis et al.* (*2000*) for a summary of the many factors affecting extinction risk). Rarely do metrics capture necessary information concisely and consistently across all species. But such metrics are needed for wildlife managers to monitor outcomes and effectively allocate resources.

Conservation program administrators need a small, yet strategic, number of highly informative and consistent metrics to efficiently and accurately evaluate the conservation status of each species and conservation programs as a whole. This need can be characterized in three ways:

1. *Simplicity.* A method to monitor the conservation status of hundreds or thousands of species has to be easy to implement. First, such a method has to be able to scale to a regional or national scope and hundreds or thousands of practitioners in the field. A complex method may provide more detail and data, but is more likely than a simple method to fail at scale. Second, limited budgets for conservation place real-world constraints on the complexity of what can be implemented.

2. *Sufficiency.* Two fundamental components of conservation status are a species' demography and the threats it faces (*Goble*, *2009*; *Neel et al.*, *2012*). Separating these factors is crucial because strategies for addressing threats and demographic status can differ greatly, e.g., population augmentation may improve demographic status while threats that will ultimately undo those gains continue unabated (*National Marine Fisheries Service*, *2010*). These two components are composed of sub-elements (e.g., population size, range, and structure are elements of demography), but knowing these details is not necessary to evaluating conservation status across a program. That is, while a single conservation status metric is insufficient, using more than two metrics is more costly and provides more detail than is necessary for evaluating a conservation program as a whole.

3. *Consistency.* Large conservation programs may cover a variety of taxa, from lichens to mammals, that are managed and monitored by hundreds or thousands of people. Suitable monitoring metrics need to be calculable and interpretable consistently across species and personnel for programmatic evaluations to be meaningful.

These characteristics point to the need for a small number of monitoring metrics that would (a) be easy to calculate given existing data, rather than requiring new and expensive research or monitoring programs, (b) capture the status or change of threats and demography independently, and (c) be designed to apply consistently across all or

most covered species. If such monitoring metrics are available, then the effectiveness of conservation programs can be evaluated in part (e.g., by geographic region) or in whole by analyzing the scores for all species under the program. For example, we could answer questions such as: What is the status of threats across all imperiled species covered by a specific conservation program? What proportion of imperiled species are declining or improving demographically? Are some geographic regions doing better, on average, at addressing the threats to imperiled species than other regions?

The U.S. Fish and Wildlife Service (FWS) reports in their Biennial Report to Congress two possible conservation status metrics for species listed under the U.S. Endangered Species Act (ESA). The first metric has changed over the years. Until 2010, FWS reported species status using categories including "declining", "improving", "stable", or "unknown." FWS stopped reporting each species' "status" after 2010 because they judged the conclusions were not scientifically rigorous enough (*U.S. Fish and Wildlife Service*, *2011*). Today, FWS reports recommendations to reclassify a species' legal status that are based on five-year reviews of each species. Recommendations may include uplisting from threatened to endangered, downlisting from endangered threatened, delisting a species, or no status change (see Article S1 for an overview of the ESA listing lifecycle). The second reported metric is the Recovery Priority Number (RPN), which is used to prioritize recovery planning for ESA-listed species. RPNs are based on the immediacy of threats, recovery potential, taxonomic uniqueness, and conflict with human activities (*U.S. Fish and Wildlife Service*, *1983*). They are represented by a "base" number from 1–18 (highest to lowest priority), and are tagged with a "C" if there is conflict with economic activity. Thus, both legal status and RPNs contain some information about conservation status and both are used in some fashion by FWS to allocate resources and make other decisions. But the question remains whether these reported metrics are acceptable for monitoring the conservation status of species, or evaluating the effectiveness of the Endangered Species program.

There are three challenges with using the metrics reported by FWS as conservation status metrics. First, a species listed as endangered cannot be afforded more protection under the ESA, and neither Congress nor the public receives an early warning if an endangered species has continued to decline. In contrast to IUCN Red List categories that include "critically endangered" and "extinct in the wild" as options before extinction (*Rodrigues et al.*, *2006*), the ESA recognizes no classification between "endangered" and "extinct". Second, some changes in either threats or demography may not be sufficient to trigger reclassification, but are still sufficient to warrant the attention of managers during the monitoring and evaluation stages of the recovery and resource allocation process. FWS administrators will be hard-pressed to make informed resource allocation decisions across the endangered species program without simple, sufficient, and consistent metrics of conservation status as part of the equation. Thus, on the first and second counts, recommendations for reclassification have significant shortcomings. Third, although used in conjunction with other information to guide resource allocation (*U.S. Fish and Wildlife Service*, *2013*), RPNs are not sufficient for evaluating species status because they combine many factors, including some that are not conditional on changes of status (e.g., taxonomic uniqueness). Because the conservation status of individual species and groups of species is the ultimate metric

by which conservation programs need to be evaluated, neither Congress, the Executive Branch, nor the public can accurately evaluate the effectiveness of the ESA at recovering species using currently reported metrics. Furthermore, some species can "fall through the cracks" of conservation while recovery progress for other species goes unacknowledged. This is not to say that such species receive no attention; biologists and managers in the field may be aware of a species' plight. But regional- or national-level administrators are much less likely to know about these issues, and cannot make informed, high-level resource allocation decisions if unaware of the facts.

Here we report on a simple, sufficient, and consistent key that can be used to translate information in detailed status reviews for imperiled species into scores denoting changes in threats and demography. This key provides the type of guidance that the Inspector General of the US Department of the Interior recommended in 2003 (*U.S. Department of the Interior IG, 2003*). To illustrate its use, we apply the key to two scenarios. First, we apply the key to 37 species for which FWS has recently recommended reclassification and 15 species without such a recommendation. While the recommendations are largely consistent with the scores extracted using the key, we confirm that they oversimplify conservation status. Specifically, recommendations of no status change are particularly prone to masking changes in the threat or demographic status of species. We also show that RPNs are moderately correlated with threat and demographic changes, but not consistent with either. Second, to illustrate its use across a specific region, we apply the key to non-plant ESA-listed species over which FWS has primary jurisdiction in Florida. The results indicate the threat and demographic statuses of these species are mostly declining, and the repeatability test indicates the need for detailed training of those who generate the scores. Building from these results, we provide recommendations for implementing the proposed scoring key for FWS's Endangered Species program that will improve the monitoring and recovery of species.

## MATERIALS & METHODS

### Key development

To develop the criteria for scoring threat and demographic changes, we studied a numerous but unrecorded number of five-year status reviews and *Federal Register* final listing rules. This review provided an overview of how the status of threats to species and their demography is discussed. To score each species for threat and demographic changes, we established a key to translate the prose of five-year reviews and Federal Register documents into scores that range from $-1$ (all or most conditions deteriorating) to $+1$ (all or most conditions improving) in increments of 0.5 (Table 1). These scores are subject to some variation in the interpretation of information in status review documents, but we have clarified the criteria for each score category to minimize the variation. Note that the criteria and scores are not the absolute status of the threats or demography of each species, but the *change* in threats or demography since each species' last review.

We applied the scoring key to two scenarios. First we evaluated whether FWS's current conservation status metrics are sufficient by focusing on species that have recently been recommended for a listing reclassification versus a sample of species that have not. We

**Table 1  The table we used to translate changes in threats and demography to quantitative scores for the evaluated species.**

| Category | Criteria | Score | | | | | |
|---|---|---|---|---|---|---|---|
| | | −1 | −0.5 | 0 | 0.5 | 1 | U |
| Threats | Most or all threats increased or impossible to address | X | | | | | |
| | Primary threats increased but others eliminated | | X | | | | |
| | Most or all threats continued unabated (no change) | | | X | | | |
| | Primary threats decreased but others increased | | | | X | | |
| | Most or all threats decreased or eliminated | | | | | X | |
| Demography | Most or all populations increased | | | | | X | |
| | Most populations increased but others decreased or eliminated | | | | X | | |
| | Most or all populations remained stable | | | X | | | |
| | Most populations decreased but others increased | | X | | | | |
| | All populations decreased | X | | | | | |
| Either | No information available | | | | | | X |

analyzed the status of all non-plant species that were recommended for reclassification by FWS in their 2011–2012 Report to Congress (*U.S. Fish and Wildlife Service*, *2013*), as well as species the Service subsequently recommended for reclassification through March 15, 2015. We searched the Federal Register and the Office of Information and Regulatory Affairs websites to identify the species proposed for reclassification since the 2011–2012 Report to Congress. In addition to the 37 species recommended for reclassification, we randomly selected 15 control species (ten endangered and five threatened) from the 2011–2012 Report to Congress that were not recommended for reclassification. Only 15 "no-change" species were chosen as controls as a compromise between the small number of species recommended for uplisting and the larger number recommended for down- or de-listing. We recognize that this approach comes at the cost of some precision that would have been afforded by a larger sample of controls. One of us (WMW) then read the most recent five-year review or Federal Register document with status information for each species and assigned a score for the change in threats and a score for the change in demography. All authors read the relevant documents and decided on a score if the appropriate score was ambiguous for a species.

The second analysis concerned the utility of scoring threat and demographic changes across many species in a region, including inferences that might be drawn and the consistency among different scorers. One of us (JWM) scored all non-plant ESA-listed species in Florida for which FWS has primary jurisdiction. Next, we selected a random sample of ten species (Table S1) and four additional people independently recorded threat and demographic change scores. Each individual works in the field of conservation, but none is an expert on the species and only one (Y-WL) was very familiar with the variety of status review documents. All individuals were provided a guidance document prior to scoring (Article S2) that mimicked what might be provided to FWS biologists before they are asked to record scores for a species. Scoring by all participants was based on the same set of Federal Register or five-year review documents for these ten species.

## Statistical analyses

The data for all species and across both analyses are available from figshare at https://dx.doi.org/10.6084/m9.figshare.3436763.v1. The data and code for this work are available as an R package at https://github.com/Defenders-ESC/threatdemog/ and can be installed using devtools::install_github("Defenders-ESC/threatdemog"). The analyses are provided in two R package vignettes, and are described below.

*Analysis 1:* we compared threat and demographic change scores to FWS's status change recommendations and to the RPNs. For all comparisons we considered four basic models:

Model 1: response ~ combined score + error
Model 2: response ~ threats + demography + error
Model 3: response ~ threats + error
Model 4: response ~ demography + error

where 'response' is either FWS's status change recommendation or the RPN for each species, and 'combined score' is simply the sum of the threats and demography change scores. Within analyses for which the calculations are feasible, we used Akaike's Information Criterion corrected for small sample sizes ($AIC_C$) for multimodel comparisons and model selection (*Burnham & Anderson*, *2002*) using the AICcmodavg package (*Mazerolle*, *2015*) for R 3.1.2 (*R Core Team*, *2015*).

We used two methods to determine if our scores were consistent with FWS status change recommendations. We first used a multinomial model, implemented with the nnet package (*Ripley & Venables*, *2015*) and with no-change as the reference class, to test if the scores for species differed by recommendation for status change. We then used discriminant function analysis from the MASS package (*Ripley et al.*, *2015*) to classify species into improving, declining, and no-change status using the four models above. We separate misclassifications into two groups: underprotection cases, in which threats and/or demography may indicate a species should have been granted additional protection, if available; and overprotection cases, in which the scores indicate the current level of protection may be unnecessarily high.

To assess whether RPNs reflect the threats and demographic change scores, we first used the four base models above with the base number and the conflict tag of the RPN as the response variables in a MANOVA (*Scheiner & Gurevitch*, *1998*). Threat and demographic change scores were not predictive of the conflict tag (all $p \gg 0.05$), so we dropped the conflict tag from further analysis. Because linear model residuals were non-normal, we used a generalized linear model with a negative binomial error distribution and log link function (*McCullagh & Nelder*, *1989*). We checked plots of residuals versus predicted values to ensure model suitability.

*Analysis 2:* we calculated summary statistics of the threats and demographic change scores, using the pastecs R package, to describe the overall status of across all scored Florida species. To evaluate the importance of inter-individual scoring variation, we used a two-way ANOVA with species and observer effects on threat and demographic scores. We followed this by calculating the intraclass correlation (on species) using the ICC package for R. Demographic and threat scores were evaluated independently, and residuals from the linear models were checked for normality.

## RESULTS

*Analysis 1:* we identified 52 species across nine taxonomic groups that met our criteria for the first analysis (summarized in Table S1). In Table S2 we provide example text from selected five-year status reviews that illustrates the range of threat and demographic scores given the criteria in Table 1. The mean threat and demographic change scores were positive across all species ($\bar{x}_{\text{threats}} = 0.173$ and $\bar{x}_{\text{demography}} = 0.038$), but there was considerable variation overall ($\text{IQR}_{\text{threats}} = 0\text{--}0.625$; $\text{IQR}_{\text{demography}} = -1\text{--}1$).

We identified 27 species recommended for downlisting or delisting, i.e., improving status. Only 15 of the 27 species had positive scores for both threats and demographics. Four had threat alleviation scores of zero (i.e., no change) but positive demographics scores. Another four had demographic scores of zero but positive threat alleviation scores. No species in this category had a negative score in both categories.

Ten species were recommended for uplisting (i.e., declining), of which only copperbelly water snake (*Nerodia erythrogaster neglecta*) had non-negative scores for both threats and demographics. All other declining species in this category had a negative demographics score and a threat alleviation score of zero or lower.

Two of the 15 control species had negative scores for both threat alleviation and demographics, and six species had a negative score for either threat alleviation or demographics. Three control species had a score of 0.5 (i.e., moderate improvement) for either threat or demographic change, and one control species, Yellow-shouldered Blackbird (*Agelaius xanthomus*), had positive scores for both threat alleviation and demographics.

Status change recommendations tended to reflect the changes in the combined threat change and demographic change scores (Fig. 1A), but the consistency with threat change scores (Fig. 1B) was weaker than the consistency with demographic change scores (Fig. 1C). This was supported by the results of the multinomial model (Table 2). Models 1 and 2 had very similar, low $\text{AIC}_C$ scores even though model 2 had twice as many parameters. Even though the $\text{AIC}_C$ for model 1 is lower, model 2 should be preferred for application because threat and demographic parameters are separable.

The results of the linear discriminant function analysis were similar to the multinomial model analysis. FWS status change recommendations were consistent with classifications from Model 2 for 81% of species, with >88% of improving and declining species consistently classified (Fig. 2). FWS status change recommendations may have resulted in underprotection of seven species and overprotection of three species given the classifications based on threat and demographic change scores. The consistency of FWS status change recommendations was 71% under Model 1 (demography and threats scores added), 65% under Model 3 (threats only), and 62% under Model 4 (demography only).

Both threat and demographic change scores were significantly correlated with FWS's RPNs (range of $r$: 0.31–0.45) at $\alpha < 0.05$ (Fig. 3). The $\text{AIC}_C$ values for Models 1–3 were within 1 unit of each other, indicating these models are approximately equally parsimonious (Table 3). Model 2, which treats threat and demographic change scores separately, had the best overall fit among the three models. However, the relative importance of threat
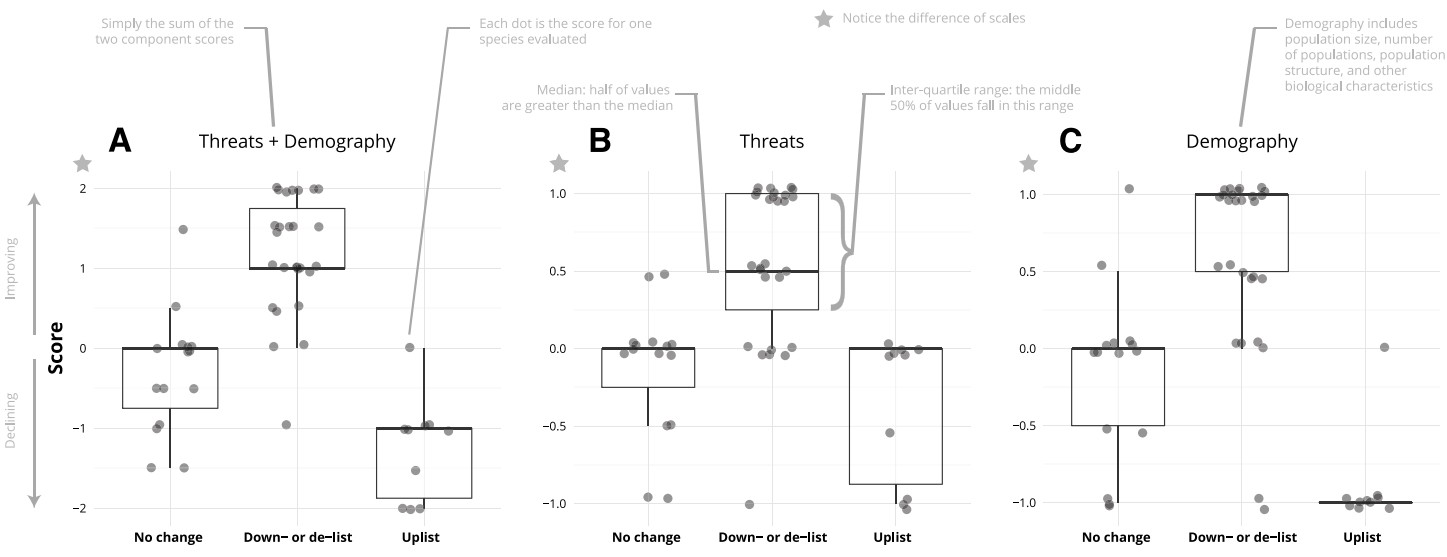

**Figure 1** **The combined threat and demography scores were generally consistent with Fish and Wildlife Service status change recommendations (A), but that pattern is weaker for threat changes (B) than for demographic changes (C).** A significant result is that seven of the 15 no-change species have negative summed scores: although no change was recommended, the species are declining in terms of threats and/or demography.

**Table 2** **Both the combined and separate threat and demography scores are significantly (at $\alpha < 0.001$) higher for improving species than for no-change species, but not significantly lower for declining species.**

| Model | Recommend.[a] | Coeff.[b] | Std. err.[b] | z-score[b] | p-value[b] | Deviance | AIC$_C$ |
|---|---|---|---|---|---|---|---|
| 1 | Uplist | −0.830 | 0.53 | −1.429 | 0.153 | 78.295 | 82.295 |
| | Down/de-list | 2.400 | 0.672 | 3.613 | 0.0003 | | |
| 2 | Uplist | −0.070/−0.877 | 0.787/0.570 | −0.089/−1.538 | 0.929/0.124 | 63.527 | 71.527 |
| | Down/de-list | 2.990/1.849 | 1.063/0.782 | 2.814/2.366 | 0.005/0.018 | | |
| 3 | Uplist | −0.386 | 0.724 | −0.533 | 0.594 | 79.894 | 83.894 |
| | Down/de-list | 3.408 | 0.967 | 3.526 | 0.0004 | | |
| 4 | Uplist | −0.830 | 0.53 | −1.568 | 0.117 | 79.295 | 82.295 |
| | Down/de-list | 2.400 | 0.672 | 3.568 | 0.0003 | | |

**Notes.**
[a]Classes of Fish and Wildlife Service status change recommendations; "no-change" is set as the reference level.
[b]Number preceding the slash is for the threat score, number after the slash is for demography score.

change scores to predicting RPNs is supported by both the higher confidence of the threat parameter estimate of Model 2 and the good fit of Model 3.

*Analysis 2:* on average, we found negative threat and demography scores across 54 ESA-listed, non-plant species in Florida over which FWS has primary jurisdiction (Table 4). Three species had positive scores for both threat and demographic status; seven had positive scores for either threat or demographic status; and the remaining 44 species had

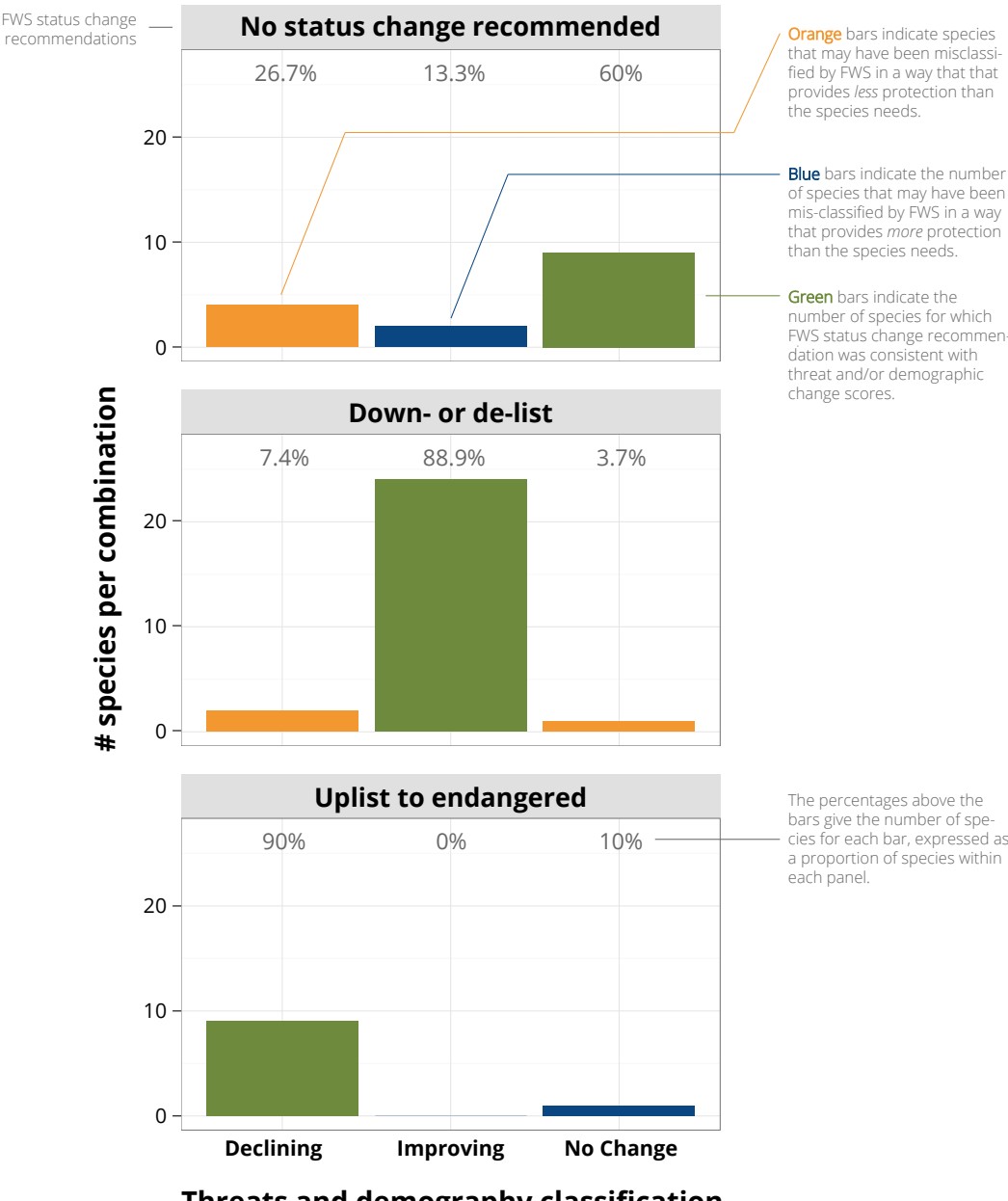

FWS status change recommendations

**No status change recommended**

26.7%  13.3%  60%

Orange bars indicate species that may have been misclassified by FWS in a way that that provides *less* protection than the species needs.

Blue bars indicate the number of species that may have been mis-classified by FWS in a way that provides *more* protection than the species needs.

Green bars indicate the number of species for which FWS status change recommendation was consistent with threat and/or demographic change scores.

**Down- or de-list**

7.4%  88.9%  3.7%

# species per combination

**Uplist to endangered**

90%  0%  10%

The percentages above the bars give the number of species for each bar, expressed as a proportion of species within each panel.

Declining  Improving  No Change

**Threats and demography classification**

**Figure 2** **While most of the Fish and Wildlife Service's (FWS) status change recommendations were consistent with threat and demographic scores, some recommendations may confer under- or over-protection.** The results from Model 2 are shown because FWS's status change recommendations had the highest consistency (81%) with the linear discriminant function analysis (LDA) of threat + demographic score. As discussed in the main text, some reclassifications that contradict the scores are explained in the detailed five-year reviews.

scores of zero or lower for both status components. Three species were recommended for downlisting and were a part of Analysis 1, 37 species were recommended for no status change, and the remaining 14 species were listed too recently for a recommendation. Threat and demography scores were correlated across all species at $r = 0.72$ ($t = 7.19, p = 3.74e^{-9}$).
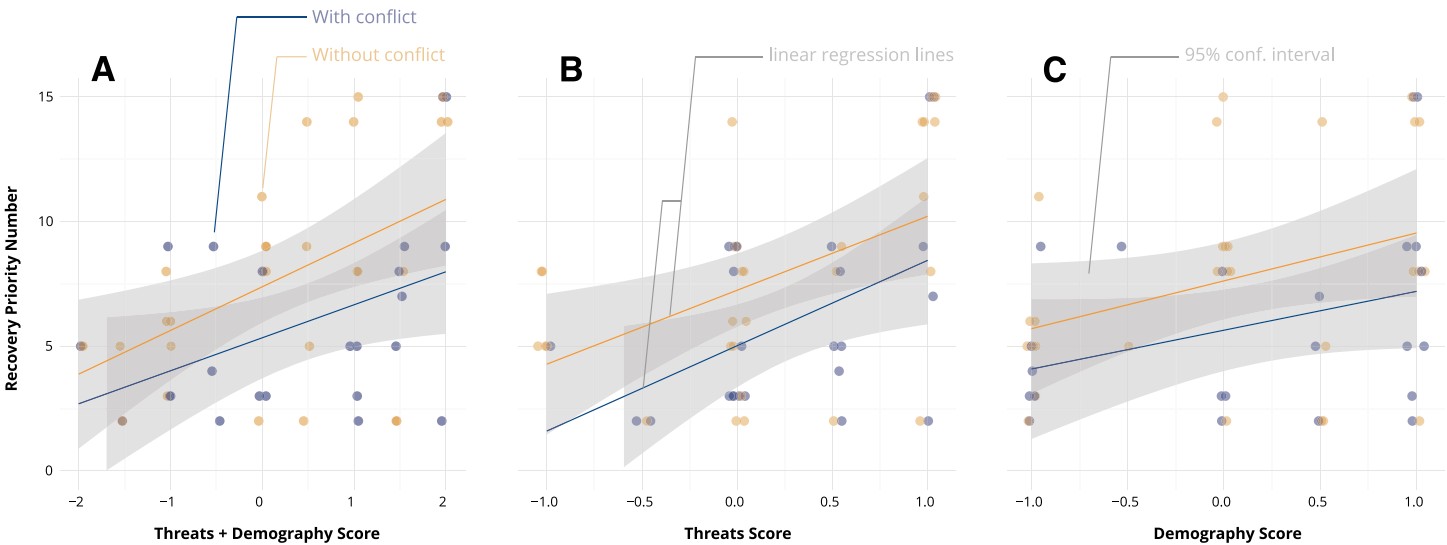

**Figure 3** Fish and Wildlife Service's Recovery Priority Numbers were correlated with the combined threat and demography scores (A, $r = 0.43$), but the correlation was stronger with threat change scores (B, $r = 0.447$) than with demography scores (C, $r = 0.31$). That is, although there is consistency between RPNs and threat and demography scores, the relationship is not particularly strong.

**Table 3** Model results for predicting Fish and Wildlife Service's Recovery Priority Number from threat and demography scores.

| Model | df | Deviance | $-2 \log L$ | $R^2$ | $AIC_C$ | Parameter | Estimate | s.e. | z-value | p-value |
|---|---|---|---|---|---|---|---|---|---|---|
| 1 | 1, 50 | 52.74 | −274.47 | 0.184 | 280.97 | Combined score | 0.215 | 0.064 | 3.36 | 0.0007 |
| 2 | 2, 49 | 53.28 | −273.08 | 0.206 | 281.93 | Threat | 0.355 | 0.139 | 2.55 | 0.011 |
|  |  |  |  |  |  | Demography | 0.109 | 0.109 | 1 | 0.316 |
| 3 | 1, 50 | 53.6 | −274.09 | 0.19 | 280.59 | Threat | 0.416 | 0.123 | 3.37 | 0.0007 |
| 4 | 1, 50 | 52.59 | −279.77 | 0.097 | 286.27 | Demography | 0.243 | 0.103 | 2.37 | 0.0177 |

**Table 4** Summary statistics for threat and demography scores of non-plant ESA-listed species in Florida over which the US Fish and Wildlife Service has primary jurisdiction.

|  | n | N/A | Min | Max | 25% | 75% | Median | Mean | s.d. |
|---|---|---|---|---|---|---|---|---|---|
| Demography | 50 | 4 | −1 | 1 | −1 | −0.125 | −1 | −0.550 | 0.702 |
| Threat | 54 | 0 | −1 | 1 | −1 | −0.5 | −1 | −0.630 | 0.525 |

We found variation in both threat and demography scores recorded by different scorers (Fig. 4). For threat scores, the identity of the person scoring was not statistically significant ($F = 0.53$, $p = 0.71$), the mean square for the person term was small (0.13) relative to the mean square for species (1.78), and the intra-species score correlation was 0.56 (95% CI [0.30–0.84]). For demography, the person term was statistically significant ($F = 4.26$, $p = 0.008$), but the mean square term was still small (0.73) relative to the mean square for species (2.34) and the intra-species score correlation was 0.70 (95% CI [0.44–0.90]). The 95% confidence interval for mean threat score across scorers did not overlap zero (−0.47 to −0.06), but did for demography score (−0.35–0.16). For comparison, the confidence intervals for both threat and demography status overlapped

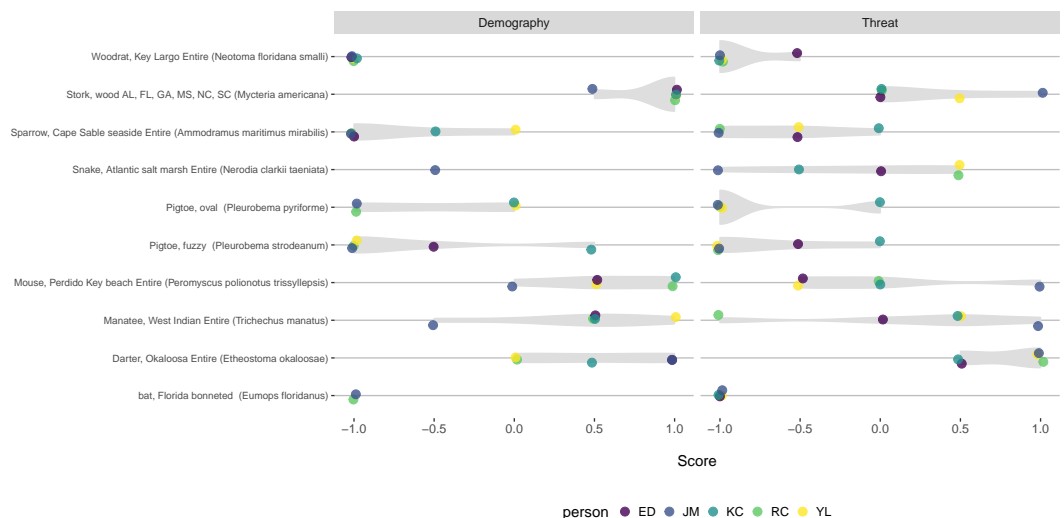

**Figure 4 Variation in threat and demography scores from five independent scorers, for ten Florida species.** While scores were fully consistent in a few cases, we observed variation in how different individuals scored threat and demography status for most species. The person recording scores was not statistically significant for threat scores ($p = 0.71$) but was for demography scores ($p = 0.008$). Despite the variation in absolute scores, the intra-species correlations were high (threat = 0.58, demography = 0.70), indicating relative consistency among scorers.

zero for these ten species when calculated from scores of the single scorer for all Florida species (JWM).

## DISCUSSION

Simple, sufficient, and consistent conservation status metrics are needed to ensure that managers of large-scale conservation programs can make informed decisions (*Kleiman et al.*, *2000*; *Ferraro & Pattanayak*, *2006*). While useful on a case-by-case basis, voluminous unstructured information (e.g., all of the data in all five-year reviews) cannot be used to evaluate the performance of a large conservation program. Too little information in status metrics can lead to unintentional neglect or poor, uninformed decisions. Simple but sufficient status metrics and quantitative summaries across species are also needed for oversight by lawmakers and for the public to understand program importance and effectiveness (*Sanderson*, *2002*; *U.S. Department of the Interior IG*, *2003*). The ESA is widely considered the strongest law in the world for imperiled species conservation (*Bean & Rowland*, *1997*), but currently no simple and sufficient conservation status metrics are reported for the species it protects. We developed a simple, sufficient, and consistent key for translating detailed conservation status information into two scores—one for threat changes and one for demographic changes—that can enable the necessary species and program evaluations. We undertook two separate analyses to illustrate the use of this key. First, while we found that changes in threats and demographics were often consistent with two metrics reported by FWS for a set of 52 ESA-listed species, our results illustrate the need for refined metrics. Second, we found that ESA-listed species from Florida are, on average, not faring well, and that there is room to improve the consistency of scoring threat

and demographic status among individual scorers. These two examples highlight applying the main result of the paper: a simple, sufficient, and (mostly) consistent key to score threat and demographic status of imperiled species at broad scales.

Our first analysis highlights a significant problem with current status reporting methods: species not recommended for listing reclassification were most likely to be neglected because of oversimplification leading to a perception of less need for attention. Forty percent of control species (no status change) showed threat and/or demography status changes even though FWS did not recommend reclassification. Because there are ca. 690 domestic, non-plant ESA-listed species, of which the vast majority are not proposed for a status change, our results suggest that a significant number may currently be treated as "stable" but are actually declining or improving. Additional data are needed to arrive at a robust estimate of the number of mis-represented species, or all species may be scored to have a census. Seven of the 15 species for which FWS recommended no status change had negative scores for threats and/or demography. Five of the seven are endangered, highlighting how FWS's current reporting metric *cannot* indicate that the status of these species is deteriorating, at least not until they are dangerously close to extinction. In contrast to the problem of masking continuing declines, we also found that improvements can be masked: four endangered species had positive scores for threat and/or demographic changes. In each of these categories, the FWS recommendation of not reclassifying hides underlying threat or demographic changes that can shape how scarce conservation resources are allocated. Future work is needed to determine the extent to which ESA-listed plants and foreign species are susceptible to this same problem.

Species recommended for reclassification are not necessarily immune from conservation neglect. While most recommended changes were consistent with the threat and demography scores, there were many inconsistencies and hidden problems. For example, three species recommended for downlisting had either a negative threat or demographic change score. Conversely, the copperbelly water snake (recommended for uplisting) had a score of zero for both threats and demography. That is, the scores for these four species reveal changes that were masked by the recommendations made by FWS. In each case, however, the authors of the reviews directly addressed the discordance between the results of the review and the recommendation. For example, the biologists for both Smith's blue butterfly (*Euphilotes enoptes smithi*) and Stephen's kangaroo rat (*Dipodomys stephensi*) concluded that threats had been ameliorated to such a degree that neither species was in imminent danger of extinction (*FWS*, *2006*; *FWS*, *2011*). In both instances, FWS reported that population size was hard to quantify because of inadequate data, but that some populations of each species were likely declining. However, FWS concluded that while threats were still present and may have contributed to lower numbers, those threats had been ameliorated or eliminated to the point that the species no longer qualify as endangered. In the case of the copperbelly water snake, the species had met uplisting criteria by the time the recovery plan was finalized (FWS 2008). At that time FWS recommended uplisting, which was recapitulated in the subsequent five-year review (*FWS*, *2010*). Cases like that of the water snake, a species that should be classified as endangered but whose scores were zero, may appear to indicate that our proposed scoring system is insufficient; how would an administrator know that the

species needs attention? We suggest that the snake might have received earlier conservation intervention if a sufficient scoring system had been in place to highlight the species' decline. Importantly, the cases above illustrate that the proposed scoring system complements, but does not replace, detailed status reviews: two scores cannot capture all of the nuances of individual species. The details may indicate a positive threat score has been, in the words of one reviewer, a threat reduction "only from really horrible to clearly unsustainable".

Our results indicate that FWS tended to use a combination of threat and demographic information in status change decisions across the analyzed species. It is unclear whether both components of conservation status are used for every species, yet both are needed to distinguish between threatened and endangered species (*U.S. Fish and Wildlife Service*, *2010*). For example, five of the ten species recommended for uplisting had negative demographic scores but threat scores of zero; none of the ten species had negative threat scores *and* demographic scores of zero. If threat changes were given more (perhaps equal) weight in uplisting decisions, then we might expect more species recommended for uplisting would have negative threat scores. The observed pattern suggests that FWS may not recommend species for uplisting until data show diminished demographic status, which may preclude early actions that can help species avoid deeper declines. In an ideal world, mechanistic models linking threats to demographic changes would be available for all species, and decisions could be made based on those threats, before symptoms arise. However, such models exist for very few species and best professional judgment must be used. We recommend that FWS biologists critically evaluate whether they are giving due weight to threats when evaluating status changes.

Our second set of analyses, focused on ESA-listed, non-plant species in Florida over which FWS has primary jurisdiction, resulted in two insights. First, we found that the 54 species had negative scores for both threat and demographic status on average, and only 18% exhibited at least one status metric that was positive. Twenty-eight species (52%) were scored $-1$ for both threat and demographic status. Simply knowing this distribution of status scores may lead administrators to investigate questions about whether there are common characteristics of species (or their management) that have at least one status component improving. Can we allocate some resources from species that are improving to species that are not, without losing the gains? Although using threat status alone may lead to suboptimal resource allocation decisions (*Joseph et al.*, *2008*), the data gained from applying the proposed key are one critical type of information needed to improve budgeting for ESA implementation (*Gerber*, *2016*).

Second, we found variation in how different individuals score threat status and demographic status, even when working from the same information. This is not unexpected given what is known about variation in expert elicitation (*Morgan*, *2014*). The lack of a statistically significant effect for scorer for the threat component, and the high intra-species correlations (i.e., consistent relative scoring among scorers) for both components, are promising results. However, the significant effect of scorer for demographic status suggests that refinement is needed. We suspect that additional training materials would improve scoring consistency, but further investigation is beyond the scope of this paper. We also note that, if the key is applied by FWS, then it will be applied by experts who
have intimate knowledge of the species in question. Our primary concern is that their additional knowledge may lead to scores based more on intuition than on critical analysis of available information in light of the criteria laid out in the scoring key. While more intensive elicitation methods may reduce inter-individual variation and unconscious biases (*Morgan*, *2014*), the limited resources for implementing this scoring key may preclude using such methods.

The proposed scoring system can substantially improve monitoring and implementation of complex imperiled species programs by enabling approaches that are currently unavailable. For example, negative threat or demographic change scores that persist over several evaluation periods should highlight species that require attention and allow conservation managers a more precise understanding of a species' conservation status. Over an extended timeframe, the change scores for a suite of species may provide data needed to warn of sudden state changes to the complex systems (*Scheffer et al.*, *2009*) of which ESA listed species are a part. Another extension of the scores is that sudden deviations from past scores can signal the need for prompt intervention. For example, the status of both Indiana bat (*Myotis sodalis*) and gray bat (*Myotis grisescens*) was reported as improving in each species' five-year review (*FWS*, *2009a*; *FWS*, *2009b*). Those reviews coincided with the appearance of white-nose syndrome (WNS; and its agent, *Pseudogymnoascus destructans*) but little was known about the potential demographic effects of the new threat (*Blehert et al.*, *2009*). The sudden appearance of a negative threat change score would stand out among the positive threat and demographic change scores for previous reviews for these species. For example, during the run-up to the ESA listing of the closely related northern long-eared bat (*Myotis septentrionalis*) in 2013–2014, the wind-energy industry proposed funding research to find pharmacological or other mechanisms (e.g., rapid adaptation and evolutionary rescue; see *Yoshida et al.*, *2007*; *Ellner, Geber & Hairston Jr*, *2011*; *Maslo & Fefferman*, *2015*) to combat WNS. If the severity of the issue had been properly flagged in 2008, then it may have triggered more effort and interest by administrators to prioritize critical research earlier. If the research was successful in identifying treatments or resistance alleles, it may have been possible to reduce the dramatic demographic declines of these species. Having quantitative threat and demographic change scores is necessary to enable these or other analyses and resultant management decisions.

Is there an alternative to the proposed key that still satisfies the needs of scoring conservation status in a way that informs program administration and monitoring? Perhaps. One reviewer suggested that scoring in 0.5-unit increments was too fine-scale; the inter-individual scoring variation we observed may support that. However, we have found the 0.5-unit increments are needed in many cases where the narrative indicates a situation other than all-or-none, and think increased training can reduce inter-individual scoring variation. While much of the focus here is on one of the five scores for threats and demography, the inclusion of the "No information available" score in the key has been very useful to FWS administrators who didn't know what *wasn't* known. The lack of a standardized way of communicating data gaps is a key problem identified in the 2003 Inspector General's report (*U.S. Department of the Interior IG*, *2003*) and addressed here. Even using a three-level system (−1, 0, 1) plus an "Unknown" category, while keeping

threats and demographic changes separate, would be amenable to statistical analysis and add critical information needed for informed decision-making.

## CONCLUSION

The proposed scoring system provides a simple, sufficient, and consistent way to track threat and demographic changes across the many taxa covered by large imperiled species programs. While scores are generated on a per-species basis, we do not intend for these scores to inappropriately be the basis for decisions such as listing or delisting (in contrast, see *Robbins*, *2009*; *Regan et al.*, *2013*). Instead, the scores for individual species may best be seen as important indicators of problems or successes that warrant deeper investigation. We recommend that conservation programs lacking a broad monitoring program that separates threat and demographic statuses implement the one proposed here. This includes FWS's and National Marine Fisheries Service Endangered Species programs, for which current reporting falls short.

We expect that implementing our proposed system adds very little workload to reporting requirements already in place, but will provide program managers and the public with much-needed information. For example, ESA-mandated five-year status reviews already require a substantial investment to compile. Adding two lines in the summary section would not require much additional work. Similarly, updating a central database of threat and demographic change scores when there is a status change, or when a five-year review is submitted, would be highly informative. Eliciting expert opinion with proper guards for recognizing uncertainties (*Morgan*, *2014*) can ensure robust estimates of threat and demographic status.

While the mean threat and demographic change scores were marginally positive across the species in analysis 1, that sample is biased toward species recommended for down- or de-listing. Based on the negative-trending scores of the no-change species examined here, the sample of 54 species from Florida, and the work of *Male & Bean* (*2005*), we suspect that mean values would be negative if data were available for all ESA-listed species. Such a result would not be surprising given the vastly inadequate funding for endangered species in the United States (*Taylor, Suckling & Rachlinski*, *2005*; *Gratwicke, Lovejoy & Wildt*, *2012*; *Negron-Ortiz*, *2014*; *Gerber*, *2016*). Using simple and sufficient status metrics may provide the evidence that underscores the need to reverse the funding shortfalls and thereby improve conservation outcomes.

## ACKNOWLEDGEMENTS

We thank numerous Fish and Wildlife Service biologists who provided hard-to-find documents for some of the species we reviewed. R Covington, E Delgado, and K Stover kindly assisted with the score repeatability experiment. J Rappaport Clark, D Barry, and N Gloman engaged in helpful discussions that improved this project. We also thank three anonymous reviewers whose feedback improved this article, and a signing reviewer, V Meretsky, whose critical review was particularly helpful.

### Funding

The authors received no funding for this work.

### Competing Interests

The authors declare there are no competing interests.

### Author Contributions

- Jacob W. Malcom conceived and designed the experiments, performed the experiments, analyzed the data, contributed reagents/materials/analysis tools, wrote the paper, prepared figures and/or tables, reviewed drafts of the paper.
- Whitney M. Webber conceived and designed the experiments, performed the experiments, analyzed the data, wrote the paper, prepared figures and/or tables, reviewed drafts of the paper.
- Ya-Wei Li conceived and designed the experiments, performed the experiments, wrote the paper, prepared figures and/or tables, reviewed drafts of the paper.

### Data Availability

Data and code at GitHub: https://github.com/Defenders-ESC/threatdemog.

Data also at figshare: https://figshare.com/articles/Imperiled_species_threat_and_demography_scoring/3436763.

### Supplemental Information

Supplemental information for this article can be found online at http://dx.doi.org/10.7717/peerj.2230#supplemental-information.

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
