# Peer review of "A simple, sufficient, and consistent method to score the status of threats and demography of imperiled species"

_PeerJ, doi:10.7717/peerj.2230_

## Round 0.1 · original submission · Minor Revisions

The work is well structured and well written. It needs small changes in how to express the results as well as in the discussion. Reviewer 1 suggests adding some papers that I believe can be useful to the discussion of the final results of the work.

Reviewer 1 ·

Basic reporting

The manuscript is well written and presents a strong case for adopting the tool described in the text. The structure of the manuscript appears to conform to a PeerJ template. Figures are clear and are relevant to the content of the article.

Experimental design

The manuscript describes the methods quite well.

Validity of the findings

The findings are valid.

Additional comments

The main idea I took away from this paper was that it makes sense to have a quantitative system that guides decision making in the case of ESA listings. That is, most of the listings decisions are implicitly accounting for threats and demography for the most part, however, it is difficult to parse out how those two components influence decisions. Moreover, having a quantitative system reinforces accountability. For example, decision makers would have to justify their decision not to list a species even though the quantitative assessments of "threats" and "demography" are trending negative. This seems like a straightforward contribution. I'm glad the authors acknowledge the next step of verifying that assignments by multiple individuals are consistent. This is important to do in my mind as it not only applies to one point in time but people making the assessments will change over time. Thus, it is important to ensure that each person making those quantitative assignments are trained to do so in the exact same manner.

Although I'm not extremely familiar with the listing process, I've certainly come across a number of papers that have argued for integrating different data and models in the listing decision-making process. I suggest the authors acknowledge these articles and put their own work in the context of the broader discussion about the validity of the ESA listing process. Possible references include:

Regan, T.J., Taylor, B.L., Thompson, G.G., Cochrane, J.F., Ralls, K., Runge, M.C. and Merrick, R., 2013. Testing decision rules for categorizing species’ extinction risk to help develop quantitative listing criteria for the US Endangered Species Act. Conservation Biology, 27(4), pp.821-831.

HIMES BOOR, G.K., 2014. A framework for developing objective and measurable recovery criteria for threatened and endangered species. Conservation biology, 28(1), pp.33-43.

Waples, R.S., Nammack, M., Cochrane, J.F. and Hutchings, J.A., 2013. A tale of two acts: endangered species listing practices in Canada and the United States. BioScience, 63(9), pp.723-734.

·

Basic reporting

no comment

Experimental design

no comment

Validity of the findings

The present application of the "control" results is not accompanied by a sufficient discussion of the limitations of the data set. The sample size (15) results in a wide confidence interval around the resulting proportion, and this should be mentioned.

The correlations presented are weak and this should also be stated.

Additional comments

Overall, a potentially useful pair of metrics for assessing progress in recovering imperiled species. However, the method seems most likely to be useful as a means of assessing programmatic progress, and summarizing it across various species characteristics – taxonomic groupings, regional groupings, etc. It seems much less useful for managing individual species, although it might allow agency central-office people to scan quickly for potential missed species of interest. The present discussion focuses more, and less effectively, on the less impressive usefulness of the approach for single-species management. One of the single-species examples is poorly chosen, and the case for using the metrics at the single-species level, overall, is just not that strong. The conclusion, in its opening sentence, points appropriately to the particular usefulness of the approach for higher-level evaluation of programmatic effectiveness, but this use of the metrics is not raised in the discussion at all, despite being, it seems, the stronger use.

I recommend rebalancing the discussion away from single-species use to use for programmatic review. The idea that biologists focused on managing and recovering a single species are going to learn something new by looking at what are essentially two generalized indices is not complimentary to the biologists in question. You are at pains to point out that existing reviews (5-year reviews) are well-crafted and nuanced, and are take care to justify recommendations that might superficially seem at odds with demographic and threat trends. Thus, your own findings suggest that single-species efforts are less likely to benefit from your suggested approach. In contrast, you raise several good points regarding programmatic review (e.g., lack of levels between 'endangered' and 'extinct,' lack of good annual summary data, lack of summary information regarding data gaps), and the suggested approach will be a useful tool both for relevant agencies and for partners and oversight groups such as the authors’ organization.

The following comments are more specific. I have somehow ended up with two versions with different line numbers, and so have re-formatted some comments to list the header words, in quotation marks, of the relevant paragraph.

Throughout, provide common names. Few readers want to look them up, and many will want to know. If peerj doesn’t like common names, argue with them.

Somewhere, note that although you generally found cases easy to “grade,” there will be cases that are more difficult, e.g., species with multiple primary threats showing different temporal trends, cryptic species for which demographic summaries are not credible, species for which there are scattered data. The acknowledgements section indicates that you, yourselves, although focused specifically on finding documentation to allow you to form these relatively simple metrics, had difficulty finding the relevant documents.

Also, note clearly that some species may have threats that have been clearly reduced but only from really horrible to clearly unsustainable – reclassifying the species might not be warranted under these circumstances.

A sentence in the paragraph beginning "To assess whether RPNs" is unclear. You have not previously used the terms base number or conflict tag. Change your terminology, or define.

Last sentence in Methods – most formal documents seem to capitalize Python.

First paragraph in Results – You report mean and SD here, but the data clearly are not symmetrical – the mean is much less than a single SD. As a result, mean and SD are not particularly helpful. Your figure shows box plots – medians and interquartile ranges. Why not report those statistics here, either together with the mean and SD or instead of them?

Second paragraph in Results – I recommend breaking the large paragraph into 3 smaller ones, at “we identified ten species” and at “Two of the 15 control species.”

Paragraph beginning "The results of the linear discriminant function," last sentence. It’s unclear what you mean by “consistency.” Consistency with what? Add some additional language and clarify.

Halfway through the paragraph beginning "Species not recommended for listing," you begin to run farther than your 15 control species really let you run. It’s a really small sample. You use your 65 and 35% values as if they were real. If you were to stop and put confidence intervals around these proportions (or one of them), you would be less quick to represent them this way. Be more honest with your readers.

Towards the end of the paragraph beginning "Species recommended for reclassification" – Cases _like that of_ N. e. neglecta . . .

The paragraph beginning "Our results indicated that" is where the single-species discussion gets particularly unhelpful. It’s not useful to suggest that professional biologists ignore either or both of threats status or demographic trends. It’s not useful to suggest that the two should be given equal weight – it’s not a null hypothesis worth investigating. Where is it written that these should be equal, or even that the data supporting the two metrics would be equal in their strength?
Later in the same paragraph, you criticize the FWS for not recommending species for uplisting until demographic data show declining numbers (note that data should be plural – show, not shows). The only way to be proactive on the basis of threats data is to have at least halfway decent models that link threat status to demographic trends. Biologists lack such models for most species. As a result, FWS has no credible means to suggest uplisting, and, as you do not note, but should, there will generally be political pressure, including law suits, to persuade them not to act where they do not have sound science. Of course, there will be pressure in the opposite direction, as well. The precautionary principle is not a principle of law in the US, although the courts might defer to the agency if it chose to use an argument along those lines. You have authors who can speak to this matter with the complexity it deserves, both from the standpoint of quantitative shortcomings and from the standpoint of law. Perhaps add some language to share their understanding and better represent the issue.

In the paragraph beginning "The proposed scoring system," the terms 'option' and 'possibilities' are unclear – use other or more words and be clearer.
In the same paragraph, you employ a particularly unfortunate choice of cases. There was (and still is) nothing that anyone could have done to protect these two bat species against WNS even if the FWS had pulled out all the stops and instituted emergency listings for both of them the instant WNS was noted. If you want a case in which a species might have been helped considerably by earlier action, this is not such a case. Might one of the Hawai’ian birds work better?

Last paragraph in Discussion, penultimate sentence – a word (probably is) is missing between 'gaps' and 'a.'

Figure 1. Can you get actual axes into this figure? The version I have is really difficult to read in any detail.

Figure 2 annotations. You state that the bars represent cases in which FWS may have made mistakes that provide more or less protection. But in fact, in text, you indicate that the FWS knew of these “inconsistencies” (poor choice of words, under the circumstances) and provided explanations for the situation. This figure, taken by itself, simply suggests that FWS makes mistakes a lot. Given that the sample size involved is paltry, and that the text provides a more honest representation of the situation, I recommend dropping figure 2, or reworking the annotations.

Figure 3 caption. An r of 0.447 is not much stronger than an r of 0.31 . Neither is strong, and you should indicate somewhere that neither is strong. It’s not clear whether your last sentence is telling us what the square of 0.43 is, or whether combined means summed, and the last sentence is providing a different result, using R^2 rather than the r reported for the other 3 results. If the former, then perhaps present R^2 immediately after the r value, and drop the last sentence. If the latter, then be clearer.
The annotations in Figure 3a (with and without conflict) are unclear and there is no accompanying discussion in Results to support them. Add text in the caption or in Results.

---

## Round 0.2 · accepted · Accept

I think that the authors have made the suggestions included by the reviewers and the paper is ready to be published.